# Enhancing Motor Performance and Physical Fitness in Children with Developmental Coordination Disorder Through Fundamental Motor Skills Exercise

**DOI:** 10.3390/healthcare12212142

**Published:** 2024-10-28

**Authors:** Kyujin Lee

**Affiliations:** Department of Adapted Physical Activity, School of Social Integration, Hankyong National University, Pyeongtaek 17738, Gyeonggi-do, Republic of Korea; lee.kyujin@hknu.ac.kr; Tel.: +82-31-6104222; Fax: +82-31-6104225

**Keywords:** dyspraxia, exercise intervention, motor coordination, timing ability, physical fitness

## Abstract

**Background:** A lack of evidence exists regarding the effects intervention has on the motor performance, including the timing ability and health-related physical fitness, of children with developmental coordination disorder (DCD). **Objectives:** We aimed to assess the effects of school-based intervention that improves fundamental motor skills (FMS) on the motor performance and health-related physical fitness of children with DCD. **Methods:** The participants were 55 children (age 8–9 years) with DCD. Children with DCD were randomly assigned to either the intervention group (*n* = 27) or control group (*n* = 28). The intervention group participated in FMS training. The control group participated in a conventional physical education class. Motor performance was evaluated before and after the intervention using the Test of Gross Motor Development, second edition; Movement Assessment Battery for Children, second edition; and the Interactive Metronome. Health-related physical fitness was assessed using the physical activity promotion system. **Results:** A significant difference was observed when we considered the interaction effect of the intervention and time regarding motor performance and health-related physical fitness; however, no significant difference was observed regarding body composition. **Conclusions:** the intervention showed significant improvements in the parameters evaluated, suggesting that a school-based intervention to improve FMS may effectively improve the motor performance and health-related physical fitness of children with DCD.

## 1. Introduction

Developmental coordination disorder (DCD) is a neurodevelopmental condition that is associated with difficulties participating in physical activity and affects approximately 5–6% of school-aged children [1,2]. Children with DCD commonly experience difficulties participating in play and sports activities due to a lack of motor competence and performance, including fundamental-motor and motor-coordination skills [3,4]. These difficulties can contribute to an avoidance of motor activity, a higher incidence of obesity-related chronic diseases, negative cardiovascular-related health outcomes, and lower levels of physical activity in adulthood [5,6,7,8,9]. Furthermore, avoidance of physical activity in school-aged children may negatively affect aspects of a child’s social inclusion and self-concept formation, potentially leading to emotional and behavioral issues [10,11]. Therefore, providing effective intervention that improves the motor performance and health-related physical fitness level of children with DCD is important.

Children with DCD have difficulties participating in physical activities and activities of daily living (ADL) [1,2]. Adequate performance of ADL relies on continuous and sequential movements with appropriate reaction times to achieve motor task completion [12,13,14]. Thus, timing ability, which is one of the motor performances, plays an important role in the development of motor skills and the achievement of a satisfactory functional performance [15]. Timing ability relies on the interactions of motor and cognitive functions with the environment and is, therefore, reduced in children with DCD [16,17]. This impairment of timing ability in children with DCD, such as uncorrected or slow reactions, can have a negative effect on their ADL [14,18,19]. Physical activity intervention is a suggested method to improve one’s timing ability [20]; however, research on the effectiveness of school-based intervention that aims to improve the timing ability for children with DCD is insufficient.

Exercise interventions for children are important because poor health-related physical fitness can have potentially serious health consequences later in life [11]. Children with DCD are less physically active and have lower levels of participation in play because they have motor functional limitations [9,21]. Previous research has suggested that an exercise program for children with DCD can have positive effects on health-related physical fitness [22]. However, differences can occur in the results of intervention depending on a child’s age and the type and periods of interventions. In fact, a clear conclusion on whether exercise intervention can improve a child’s health-related physical fitness in early childhood has been difficult to reach.

Delayed motor development persists into adolescence or even adulthood in over half of the children with DCD; therefore, appropriate early interventions should be provided [9,23]. According to previous studies, school-based interventions to improve fundamental motor skills (FMS) have shown positive effects on the proficiency of FMS, participation in physical activity, and self-perceived competence in children with DCD [24]. Although previous studies have reported improvements in the functional ability of children with DCD who received training in FMS, only a limited number of studies have been conducted that investigated the change in motor performance, including the timing ability and health-related physical fitness, of children with DCD [4,24,25]. An Interactive Metronome (IM) is designed to activate the central nervous system and evaluate children with nerve disorders that are associated with reduced cognitive and motor functions. To the best of our knowledge, no intervention study in children with DCD that utilizes training in FMS and measures timing ability with an IM has been reported. 

This study assessed the changes of motor performance, timing ability, and health-related physical fitness in children with DCD following a 12-week exercise program that targeted FMS. We hypothesized that the motor performance and health-related physical fitness of children with DCD would improve after participation in the intervention.

## 2. Methods

### 2.1. Participants

We recruited 486 participants (245 boys, 241 girls) aged 8–9 years from two different elementary schools in South Korea. Screening for DCD was based on the criteria described in the DSM-5 and relevant diagnostic tools, including MABC-2 and DCDQ’07. The two primary schools were chosen for their ability to recruit sufficient DCD-diagnosed children and their suitability for conducting the intervention. Screening for DCD was based on the four criteria described in the Diagnostic and Statistical Manual of Mental Disorders, fifth edition (DSM-5); a total score on the Movement Assessment Battery for Children, second edition (MABC-2), <67 (below 15%) [26]; Developmental Coordination Questionnaire 2007 (DCDQ’07) score < 55, based on a parental report describing the level of ADL coordination of the child [27]; difficulty in learning or fulfilling school assignments, as described by the teachers; and absence of physical or neurological deficits, as reported by the parents and confirmed by health records [28]. The participants were classified into a DCD group (*n* = 81; 46 boys, 35 girls) and a non-DCD group (*n* = 405; 199 boys, 206 girls). Only 55 children with DCD (6 high-risk DCD, 49 mild-risk DCD; 35 boys, 20 girls) agreed to undergo further measurements (motor performance and health-related physical fitness); therefore, only these 55 children were included in the final analysis (Table 1). 

All 55 children with DCD (8.59 ± 0.42 years) who were included in the final analysis lived in the city of Incheon. The children were randomly assigned to either the intervention group (*n* = 27) or the control group (*n* = 28). The baseline characteristics related to the motor coordination ability of the two DCD groups did not differ significantly (MABC-2 score, *p* = 0.921). This study protocol was approved by the university’s Institutional Review Board (approval No. 1603/001-028), and informed consent for participation was obtained from all students, parents, and relevant school officials. This investigation had no conflicts of interest. Table 1 shows the characteristics of the participants. A priori power analyses indicated that we needed to have a minimum sample size of 54 children with DCD.

### 2.2. Procedure

All children with DCD were randomly divided into two groups using a computerized random number generator: the intervention group (27 children with DCD; 17 boys, 10 girls) and the control group (28 children with DCD; 18 boys, 10 girls). The randomization process ensured that each child had an equal chance of being assigned to either group, thereby minimizing selection bias. We conducted a pre-test, intervention, and post-test. The pre- and post-test assessment procedures and methods were identical for both groups. Participants in both groups were requested to evaluate motor performance by using the Test of Gross Motor Development, second edition (TGMD-2), MABC-2, and IM for the pre- and post-tests. Children with DCD in the intervention group attended a training program in FMS (36 sessions for 12 weeks), and children with DCD in the control group took a conventional physical education class for the same duration.

### 2.3. Intervention

The intervention consisted of 60-min sessions that were conducted three times a week for 12 weeks. The program was based on promoting FMS and specifically designed to develop locomotor skills, object control skills, and balance. The locomotor skills included running, galloping, sliding, leaping, hopping, and jumping. The object control skills involved basic movements typically used in soccer, basketball, and baseball. Balance skills were comprised of walking along a line, standing on one foot, and jumping on one foot. Table 2 shows an overview of the 12-week exercise program in FMS.

The exercise program was developed by considering the effective intervention activity plan presented by the CanChild Centre for Childhood Disability Research at McMaster University. This training in FMS provided assignments of suitable levels, and four to six children trained with two physical education teachers to facilitate teacher-learner interactions. The program used various teaching materials and aids to create an environment suitable for children with DCD. A specially trained physical education teacher helped the children focus on the assignment in a positive class environment. During the study period, the control group participated in regular physical education **classes 2–3 times per week during the 12-week intervention period**. These classes included **basic physical activities such as running, jumping, and stretching**, as well **as team sports like soccer and basketball**. The control group did not receive any targeted interventions focused on fundamental motor skills (FMS), but participated in activities aligned with the standard school curriculum for physical education.

### 2.4. Measurement

#### 2.4.1. Motor Performance

Motor performance was assessed in terms of three different variables: fundamental gross motor skills, motor coordination ability, and timing ability.

Fundamental gross motor skills were assessed using the TGMD-2 which evaluates control skills involving an object (hitting, throwing, catching, kicking, dribbling, and underhand rolling of a ball) and locomotor skills (running, galloping, sliding, leaping, hopping, and jumping) [29]. Before testing, the participants were given an opportunity to watch a demonstration of each motor skill without verbal cues and subsequently performed one practice trial and two evaluation trials. The entire process was video recorded, and their motor skill performances were analyzed by three TGMD-2-certified researchers. The raw scores for object control skills and locomotor skills were converted into standardized scores. The reliability value for the TGMD-2 from the three scorers was 0.997.

Motor coordination ability was evaluated using the MABC-2 in terms of manual dexterity, ball skills, and static and dynamic balance for age band 2, which is designed for evaluating children aged 7–10 years [26]. The evaluation took approximately 30–40 min and was performed by a trained evaluator who followed the procedure described for the MABC-2. All evaluations were conducted as one-to-one interactions. The raw MABC-2 score was normalized for age and presented as a percentile. The general recommendation to establish a diagnosis of DCD is a MABC-2 score of <5% [1]; however, a cutoff of 15% is generally used in experimental and clinical research [30].

Timing ability was evaluated using the IM (IM Pro 9.0; Interactive Metronome, Sunrise, FL, USA). Timing abilities were measured by timing the use of the hand, foot, or bilateral performance during 14 tasks: (1) both hands; (2) right hand; (3) left hand; (4) both toes; (5) right toe; (6) left toe; (7) both heels; (8) right heel; (9) left heel; (10) right hand/left toe; (11) left hand/right toe; (12) balance on right foot; (13) balance on left foot; and (14) repeat both hands. According to the IM manual, timing ability is classified and evaluated as the performance in each hand, each foot, both hands simultaneously, both feet simultaneously, hand and foot on the same side of the body (unilateral performance), and hand and foot on opposite sides of the body (bilateral performance). However, the literature has significant overlap in the definitions used; therefore, we used a set of modified evaluation criteria that assessed timing ability for hand performance (tasks 1, 2, 3, and 14), foot performance (tasks 4, 5, 6, 7, 8, 9, 12, and 13), and bilateral performance (tasks 10 and 11) [17].

The participants performed each of the 14 tasks by using their hand to tap a sensor attached to their hand or thigh, or by using their foot to tap a sensor mat in response to a metronome sound. For the motor tasks included in the evaluation, the expected response time varied between 0 and 44,554 s, depending on the complexity of the task and the participant’s ability to react effectively and efficiently to the metronome sound. Better timing abilities corresponded to lower response times. The assessment was performed according to a standardized 15-session protocol, meaning that each participant completed 15 sessions of the IM assessment. This protocol ensured consistent measurement of timing abilities, with each session designed to evaluate specific aspects of motor coordination and response timing [31].

#### 2.4.2. Health-Related Physical Fitness

Cardiorespiratory fitness, muscle strength, muscle endurance, flexibility, and body fat percentage were evaluated as descriptors of health-related physical fitness. Measurements were performed using the Physical Activity Promotion System (PAPS), which is typically applied to assess physical fitness among Korean elementary school, middle school, and high school students. Specifically, schools perform obligatory PAPS-based measurements to evaluate the students’ health and physical fitness and to provide recommendations for physical activities.

### 2.5. Data Analysis

IBM SPSS Statistics 21 was utilized to analyze the data. The Kolmogorov–Smirnov test was conducted to assess the normality of the data. An independent *t*-test was conducted to confirm whether there was any significant difference between the intervention and control group before the intervention. A two-way ANOVA with repeated measures was used to investigate the effect of the intervention by using Tukey’s test for post-hoc analysis. The two main factors were time (pre- and post-tests) and group (intervention vs. control). The significant alpha level was set at 0.05.

## 3. Results

### 3.1. The Effect of Intervention on Motor Performance

The two-way ANOVA with repeated measures showed a significant improvement of motor performance in the intervention group. The indicators of motor coordination ability (aiming and catching score, *p* = 0.001; balance score, *p* = 0.004; MABC-2 total score, *p* = 0.001) and fundamental gross motor skills (locomotor score, *p* = 0.001; object control score, *p* = 0.001; TGMD-2 total score, *p* = 0.001) showed significant interaction effects between the groups and time (Table 3). Additionally, the indicators of timing ability (response time hands, *p* = 0.028; response time feet, *p* = 0.018; response time bilateral, *p* = 0.001; mean response time, *p* = 0.001) showed significant interaction effects between the groups and time (Table 4).

### 3.2. The Effect of Intervention on Health-Related Physical Fitness

The two-way ANOVA with repeated measures showed a significant improvement of cardiorespiratory fitness, muscle strength and endurance, and flexibility in the intervention group. The indicators of cardiorespiratory fitness (15-m shuttle run, *p* = 0.001), muscle strength and endurance (handgrip strength, *p* = 0.012; Sargent jump, *p* = 0.011; curl-up, *p* = 0.001), and flexibility (sit-and-reach, *p* = 0.010) showed significant interaction effects between the groups and time (Table 5). In contrast, the indicator of body composition did not show a significant difference between the intervention and control groups.

## 4. Discussion

We explored the potential of FMS training to enhance motor performance (including coordination, FMS, and timing ability) and health-related physical fitness in children with DCD. The findings suggest that targeted interventions may benefit key aspects of physical development, though certain areas like body composition might require additional focus. This points to the complexity of designing comprehensive interventions that address all facets of physical health in children with DCD.

We found that the MABC-2 score, which evaluates the motor coordination ability of children with DCD, improved after the children participated in the 12-week training (Table 3). These results are in agreement with those of previous studies that reported an improvement in coordination ability after an exercise intervention [31,32,33]. Most notably, nine out of 27 children with DCD in our study (MABC-2 score ≤ 15%) showed a reduction of symptoms to the point that the children were no longer considered to be at risk for DCD (MABC-2 score > 15%). We are unable to conclude that symptoms can fully resolve within 12 weeks of an exercise intervention because our DCD group included only three children with severe indicator symptoms (MABC-2 score ≤ 5%); however, this possibility cannot be ignored. The MABC-2 score of the bottom 5% of children improved from 52.5 to 72. Additionally, no significant change was observed in the manual dexterity score because the intervention program focused only on the improvement of FMS and did not include fine motor skill exercises.

Previous studies reported that training in FMS had positive effects on motor skills and activity in children [25,34,35]. Our results show that locomotor and object control skills improved significantly in the intervention group (Table 3). This finding suggests that the intervention program was effective at improving the FMS of children with DCD and is in agreement with previous research that showed training in FMS supported by error-reduced learning was effective in a sample of children with DCD [25]. Notably, our observed improvement of FMS is higher than in previously reported results [25,35]. Our results showed that around 51% of FMS improved and reached the normal standard range for TGMD-2. Therefore, we have demonstrated that intervention has the ability to improve the FMS of children with DCD to normal levels.

This investigation is the first to evaluate the effect of training in FMS on the timing ability of children with DCD using the IM. Our findings reveal that the intervention group showed interactive effects on timing abilities. The different components of the timing ability in children with DCD showed significant effects for all assessments: hands 23% (pre-post, 139.43 milliseconds–107.38 milliseconds), feet 25.9% (pre-post, 192.31 milliseconds–142.52 milliseconds), and bilateral 25.8% (pre-post, 210.24 milliseconds–156.04 milliseconds). Significant changes were also observed within the control group (hands 5.6%, feet 8.5%, and bilateral performance 5%), although these changes were small compared to the results of the intervention group and were most likely due to a repetition of the active metronome evaluation. This trend is consistent with the findings of previous studies that reported intervention effects in the timing ability of children with DCD [20,36].

An analysis of the IM-based test data shows that children with DCD had poor reaction times in hand, foot, and bilateral performance. The most pronounced difference was in task 11 (bilateral performance), which is consistent with the findings of Rosenblum and Regev (2013) [17]. These results demonstrate that children with DCD have difficulties reacting to auditory signals and simultaneously carrying out designated movements, and this difficulty is more pronounced for tasks involving symmetrical movement because a higher coordination ability is required. Our findings are in agreement with previous observations that suggested children with DCD have slower motor reactions to auditory stimuli [37,38]. Whitall et al., attributed the cause of this phenomenon to a deficit in auditory processing associated with DCD [38].

Children’s health-related physical fitness has important implications in relation to development [7]. Health-related physical fitness has been considered an important factor for the healthy development of children with DCD [21]. Both longitudinal and cross-sectional studies have indicated that children with DCD have poorer health-related physical fitness than children with typical development [39,40]. As previously reported, children with DCD have poor indicators of health-related physical fitness, which is consistent with our findings [39,40]. However, these indicators improved after participation in the 12-week exercise program focused on training in FMS; this improvement was most likely due to increased participation in physical activities, which would be expected to positively influence cardiorespiratory fitness, muscle strength, muscular endurance, and flexibility. However, the percentage of body fat increased significantly in the children from both the intervention group (DCD-int) and the control group (DCD-con); this increase is likely related to the fact that diet was not restricted during the study period, and the participants were going through a major developmental growth phase [41].

The positive results in the timing ability of children with DCD could be related to the content of the intervention program and the method of teaching. Children with DCD use their hands, feet, and body to improve their FMS, such as catching, hitting, throwing, and kicking, through trial and error of timing and spatial awareness. Such exercises in trial and error enabled the successful planning and subsequent execution of movement-related timing abilities. This process is similar to the changes of timing ability within the internal models [42]. The reasons behind the success of our intervention program can be found in the principles of the provision of education environments for children with DCD and the task-oriented class method [43]. Specifically, our program included physical activities tailored for children with DCD that were integrated into their regular education environment, and our program limited the number of children in each session so that each child had sufficient interaction with the teacher. Furthermore, although the nature and difficulty of the tasks were decided beforehand, the supervising teacher could modify these parameters based on the individual response of each child. By observing their own success in various tasks, children with DCD gained confidence in their motor skills, and this confidence motivated them to continue training and effectively helped them improve their motor performances [25,31]. 

Scientific results from both our study and previous studies indicate that children with DCD can improve their motor abilities and health-related physical fitness with appropriate interventions. For young children who especially need to develop their FMS, participation in physical activities involving play and sports can promote motor coordination and health-related physical fitness. We utilized a 12-week intervention program with a focus on FMS in a school setting. However, whether the motor deficit noted in children with DCD is innate or represents a lack of appropriate educational activities is unclear. Further research is needed to investigate the mechanisms of change in both motor behavior and the physiological, neurological, and cognitive aspects of motor skill development.

This study had several limitations. First, we recruited children from only two schools in Incheon City; therefore, our findings should be interpreted with caution. Second, we excluded participants with medical diagnoses other than DCD, which could result in a potential bias since the DSM-5 considers that DCD co-occurs with other conditions. However, in our case, we only excluded one child with intellectual disabilities whose school performance was difficult to evaluate by teachers. Third, this study may not sufficiently reflect the characteristics of DCD because of the small sample size (27–28 children with DCD in each group) and use of a MABC-2 cut-off standard below 15% for DCD screening. Fourth, the low teacher-student ratio in the intervention group may have had a positive effect on the results. Hence, future studies should be conducted to examine these various limitations.

## 5. Conclusions

This research aimed to clarify the effects of FMS training on the motor performance and health-related physical fitness of children with DCD. **The results demonstrated significant improvements** in motor **coordination**, timing abilities, and health-related physical fitness **among children who participated in the 12-week intervention program**. These findings suggest that the intervention may positively impact the daily lives of children with DCD enrolled in public elementary schools. Future studies should **build upon these results to further refine and optimize** exercise **interventions targeting** motor performance and health-related **outcomes in** children with DCD.

## Figures and Tables

**Table 1 healthcare-12-02142-t001:** Characteristics of the participants.

Characteristic	Intervention Group (*n* = 27)	Control Group (*n* = 28)	*p*-Value
Age (years)	8.62 ± 0.44	8.56 ± 0.39	-
Gender (boys, girls)	17, 10	18, 10	-
Height (cm)	130.48 ± 5.72	130.59 ± 4.98	0.941
Weight (kg)	30.81 ± 6.57	29.91 ± 5.81	0.848
BMI (kg/$m^{2}$)	18.08 ± 2.77	17.54 ± 2.81	0.819
Total MABC-2 score	61.59 ± 4.92	62.34 ± 5.81	0.921
Total MABC-2 (%)	10.58 ± 4.41	11.02 ± 4.71	0.901

*Note.* BMI, body mass index; MABC-2 = Movement Assessment Battery for Children, second edition.

**Table 2 healthcare-12-02142-t002:** Overview of the 12-week fundamental motor skill exercise program.

Activity (Duration)	Weeks 1–4	Weeks 5–8	Weeks 9–12
Warm-up (10 min)	▪ Rhythm activities ▪ Stretching	▪ Rhythm activities ▪ Stretching	▪ Rhythm activities ▪ Stretching
Practice to improve locomotor skills (10 min)	▪ Learn basic motor skills for -Running: Running to the hoop while swinging the arms and legs -Galloping: Passing over the hoop with a galloping step using both feet for take-off and landing -Hopping: Jumping with one foot to pass over the hoop without any assistance	▪ Practice fundamental and advanced locomotor skills ▪ Practice running, galloping, and hopping -Leaping: Running to the hoop and jumping over it with a hurdle movement -Jumping: Consecutively jumping with two feet over an obstacle -Sliding: Moving horizontally along a line	▪ Practice without obstacles -Running, galloping, hopping -Leaping, jumping, sliding ▪ Practice with obstacles: -Running, galloping, hopping -Leaping, jumping, sliding
Practice to improve object control skills (25 min)	▪ Basketball-based skills: -Rolling the basketball forward like a bowling ball -Bouncing the basketball in place and while walking without external perturbation -Chest pass (throwing and catching) with/without walking or running -Overhand pass with/without walking or running	▪ Soccer-based skills: -Kicking the stopped or moving ball with the top of the foot -Soccer dribbling in place and while walking with /without obstacles -Soccer passing and trapping with/without walking or running -Soccer game	▪ Baseball-based skills: -Overhand throw with /without a target -Holding a bat and swinging with the stopped or moving ball -Throwing and catching the ball with/without walking or running -Hitting a non-moving ball with the stopped or moving ball -Baseball game
Practice to improve balance (10 min)	▪ Walking along the line ▪ Standing on one foot ▪ Jumping on one foot (five hops) and keeping balance ▪ Keeping balance while walking back and forth	▪ Walking along the line with tiptoes ▪ Standing on two or one foot on a balance board ▪ Jumping on one foot (more than five hops) and keeping balance ▪ Keeping balance while walking back, forth, and sideways with a narrow base of support	▪ Walking on a balance beam with/without assistant ▪ Standing on one foot with eyes closed ▪ Consecutively jumping on one foot and keeping balance in various settings ▪ Keeping balance while walking back, forth, and sideways with a narrow base of support
Cooling down (5 min)	▪ Stretching	▪ Stretching	▪ Stretching

**Table 3 healthcare-12-02142-t003:** Intervention effect on motor performance in children with DCD.

Scale		Group	Pre-Test	Post-Test	Time × Group
MABC-2	Manual dexterity	Intervention (*n* = 27) Control (*n* = 28)	22.09 ± 4.42 21.61 ± 2.13	24.41 ± 2.41 22.81 ± 2.05	*F =* 1.504 *p* = 0.297
	Aiming and catching	Intervention (*n* = 27) Control (*n* = 28)	11.91 ± 2.72 11.61 ± 2.45	16.52 ± 3.16 11.81 ± 2.41	*F =* 18.841 *p* = 0.001 ✝
	Balance	Intervention (*n* = 27) Control (*n* = 28)	27.59 ± 3.81 29.12 ± 4.08	32.92 ± 3.99 28.59 ± 4.21	*F =* 10.522 *p* = 0.004 ✝
	MABC-2 total	Intervention (*n* = 27) Control (*n* = 28)	61.59 ± 4.92 62.34 ± 5.81	73.85 ± 6.82 63.21 ± 6.19	*F* = 25.184 *p =* 0.001 ✝
TGMD-2	Locomotor	Intervention (*n* = 27) Control (*n* = 28)	30.27 ± 6.52 29.21 ± 6.96	41.41 ± 3.17 29.10 ± 5.82	*F =* 42.989 *p* = 0.001 ✝
	Object control	Intervention (*n* = 27) Control (*n* = 28)	21.12 ± 7.81 19.38 ± 6.71	35.28 ± 4.89 21.28 ± 6.44	*F* = 77.287 *p* = 0.001 ✝
	TGMD-2 total	Intervention (*n* = 27) Control (*n* = 28)	51.39 ± 10.05 48.59 ± 8.89	76.69 ± 7.01 50.38 ± 8.12	*F* = 80.823 *p* = 0.001 ✝

*Note.*✝
Significant difference between the groups for the same time point; MABC-2 = Movement Assessment Battery for Children, second edition; TGMD-2 = Test of Gross Motor Development, second edition.

**Table 4 healthcare-12-02142-t004:** Intervention effect on timing ability in children with DCD.

Component	Group	Pre-Test	Post-Test	Time × Group
Mean RT hands	Intervention (*n* = 27) Control (*n* = 28)	139.43 ± 60.71 145.71 ± 79.25	107.38 ± 52.89 137.55 ± 82.24	*F* = 6.428 *p* = 0.028 ✝
Mean RT feet	Intervention (*n* = 27) Control (*n* = 28)	192.31 ± 67.38 164.18 ± 39.86	142.52 ± 60.53 150.29 ± 38.32	*F* = 6.689 *p* = 0.018 ✝
Mean RT bilateral	Intervention (*n* = 27) Control (*n* = 28)	210.24 ± 73.29 183.22 ± 69.13	156.04 ± 59.65 174.13 ± 71.34	*F* = 19.438 *p* = 0.001 ✝
Mean RT	Intervention (*n* = 27) Control (*n* = 28)	180.66 ± 66.79 164.37 ± 42.66	135.31 ± 57.63 153.99 ± 63.93	*F* = 20.132 *p* = 0.001 ✝

*Note.*✝
Significant difference between the groups for the same time point; RT = response time (millisecond).

**Table 5 healthcare-12-02142-t005:** Intervention effect on health-related physical fitness in children with DCD.

Component	Test Item	Group	Pre-Test	Post-Test	Time × Group
Cardiorespiratory fitness	15-m shuttle run	Intervention (*n* = 27) Control (*n* = 28)	19.52 ± 9.51 26.2 ± 10.13	43.13 ± 12.61 28.29 ± 8.29	*F* = 25.312 *p* = 0.001 ✝
Muscle strength and endurance	Handgrip strength	Intervention (*n* = 27) Control (*n* = 28)	9.78 ± 2.19 9.68 ± 1.97	11.72 ± 1.93 9.79 ± 1.51	*F* = 8.182 *p* = 0.012 ✝
	Leg strength	Intervention (*n* = 27) Control (*n* = 28)	35.12 ± 8.23 37.31 ± 6.89	36.12 ± 7.62 38.40 ± 6.81	*F* = 0.930 *p* = 0.498
	Sargent jump	Intervention (*n* = 27) Control (*n* = 28)	26.25 ± 4.48 27.20 ± 6.81	34.88 ± 8.62 29.02 ± 7.59	*F* = 8.061 *p* = 0.011 ✝
	Curl-up	Intervention (*n* = 27) Control (*n* = 28)	5.12 ± 4.12 7.42 ± 4.65	14.21 ± 11.03 8.02 ± 4.76	*F* = 13.125 *p* = 0.001 ✝
Flexibility	Sit and reach	Intervention (*n* = 27) Control (*n* = 28)	1.68 ± 11.29 1.56 ± 9.38	5.12 ± 8.62 1.23 ± 8.32	*F* = 8.271 *p* = 0.010 ✝
Body composition	% Fat	Intervention (*n* = 27) Control (*n* = 28)	29.52 ± 5.11 32.12 ± 4.76	31.18 ± 5.17 34.77 ± 4.18	*F* = 0.992 *p* = 0.396

*Note.*✝
Significant difference between the groups for the same time point.

## Data Availability

The data are not publicly available due to privacy or ethical concerns.

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
