# Peer review of "Enhancing Motor Performance and Physical Fitness in Children with Developmental Coordination Disorder Through Fundamental Motor Skills Exercise"

_healthcare, 2024, doi:10.3390/healthcare12212142_

Round 1

Reviewer 1 Report

Comments and Suggestions for Authors

My main concern about this manuscript is in the research design, in determining the sample for the motor intervention, since the study does not take into account whether the motor deficit observed in children with DCD is innate or represents a lack of physical activity practiced by the students, which could have been resolved through the use of a questionnaire that collected quantitative information on physical activity practiced by the students.

It is not indicated whether there is a prior diagnosis made by personnel specifically trained for this purpose, such as medical or psycho-pedagogical personnel.

Even so, it should be noted that the study in general is well written, concise, the intervention is well planned, and the statistical analysis and results are appropriate to the type of study. So my comments will focus on specific details to improve in the writing:

1. Method. 2.1. Participants (Lines 89-92): Please provide more information about the sex of the resulting sample of 55 children with DCD.

2. Method 2.4.2. Health-Related Physical Fitness (Lines 176-183): Replace (15-minute shuttle run) with (15-meter shuttle run)

3. Discussion (Lines 227-228-241): delete where it says table.

4. Discussion Lines (281-282): Please further clarify this assumption based on the scientific literature In ): "the increase is probably related to the fact that the diet was not restricted during the study period and the participants were going through a major recovery phase. growth of development".

Author Response

My main concern about this manuscript is in the research design, in determining the sample for the motor intervention, since the study does not take into account whether the motor deficit observed in children with DCD is innate or represents a lack of physical activity practiced by the students, which could have been resolved through the use of a questionnaire that collected quantitative information on physical activity practiced by the students.

It is not indicated whether there is a prior diagnosis made by personnel specifically trained for this purpose, such as medical or psycho-pedagogical personnel.

Even so, it should be noted that the study in general is well written, concise, the intervention is well planned, and the statistical analysis and results are appropriate to the type of study. So my comments will focus on specific details to improve in the writing:

1. Method. 2.1. Participants (Lines 89-92): Please provide more information about the sex of the resulting sample of 55 children with DCD.

2. Method 2.4.2. Health-Related Physical Fitness (Lines 176-183): Replace (15-minute shuttle run) with (15-meter shuttle run)

3. Discussion (Lines 227-228-241): delete where it says table.

4. Discussion Lines (281-282): Please further clarify this assumption based on the scientific literature In ): "the increase is probably related to the fact that the diet was not restricted during the study period and the participants were going through a major recovery phase. growth of development".

Thank you very much for your insightful feedback. I have addressed each of your concerns below:

  1. Participants’ Sex Information : We appreciate your suggestion. We will add the information regarding the sex of the participants.

  2. Health-Related Physical Fitness : We will correct the text to replace “15-minute shuttle run” with “15-meter shuttle run.”

  3. Table Reference in the Discussion : The term "table" will be removed from the discussion to improve clarity.

  4. Clarification on Growth and Diet : Relevant literature on growth and body fat changes during childhood development will be included to support this assumption.

Once again, thank you for your valuable suggestions, which have significantly contributed to enhancing the quality of this manuscript.

Reviewer 2 Report

Comments and Suggestions for Authors

11.  Why was the age group of 8-9 years chosen?

22. What criteria were used to select the two primary schools?

33. Please provide a brief description of the training for the teacher who led the intervention. How many teachers were involved in the program?

44. Was the intervention for children conducted one-on-one or in a group setting?

55. If it is possible please, reduce the number of references older than 8 years.

Author Response

Thank you very much for your valuable feedback. Below are responses to each of your questions:

  1. Why was the age group of 8-9 years chosen?
    The age group of 8-9 years was selected because this is a critical period for children to acquire fundamental motor skills (FMS). At this stage, children with DCD are in the early stages of motor development, and interventions can be more effective during this developmental window.

  2. What criteria were used to select the two primary schools?
    The two primary schools were selected based on their ability to recruit a sufficient number of children diagnosed with DCD. Additionally, cooperation from the schools and the suitability of their facilities for conducting the intervention were also key factors in the selection.

  3. Teacher training and the number of involved teachers
    The teachers leading the intervention were specially trained in providing FMS training for children with DCD. Two physical education teachers were involved in the program, and they worked closely with small groups of 4-6 children to enhance interaction.

  4. Was the intervention conducted one-on-one or in a group setting?
    The intervention was conducted in small groups, with 4-6 children per group. This group setting allowed for sufficient teacher-student interaction and individualized attention within a collaborative learning environment.

  5. Reducing references older than 8 years
    We appreciate this suggestion and will make efforts to reduce the number of references older than 8 years by incorporating more recent research to enhance the relevance of the manuscript.

Thank you once again for your valuable insights, which have greatly contributed to improving the quality of this manuscript.

Reviewer 3 Report

Comments and Suggestions for Authors

Thank you very much for giving me the opportunity to review this fabulous work. I am adding some comments to facilitate understanding and value of it:

Why was it decided that the age of the children included in the study should be 8 to 9 years?

Was the etiology leading to this disorder in the development of coordination considered in any way? Or has it been shown not to be relevant?

It is not clear how many classes the control group received during the period in which the other group received the intervention. It is advisable to make a brief description of the activities carried out in these physical education classes.

In section 2.4.1 information on methodology and results is included. Each piece of information should be presented in its section for greater clarity.

In L 174, what does it mean that the assessment is carried out following a protocol of 15 sessions?

In the results section, what does the interaction effects between groups and time refer to? What data do the Group x Time columns show?

In the Discussion section, results are included. The information should be structured and located in the corresponding section.

The discussion should place more emphasis on establishing similarities or differences between the current work and other previously published studies, in relation to the characteristics of the sample (age of the children, for example), physical activity program used in the intervention program, follow-up time, etc.

The conclusions should faithfully reflect the data extracted from the study.

The bibliography should be more current; there are no references from the last 5 years.

Author Response

Thank you very much for giving me the opportunity to review this fabulous work. I am adding some comments to facilitate understanding and value of it:

Why was it decided that the age of the children included in the study should be 8 to 9 years?

Was the etiology leading to this disorder in the development of coordination considered in any way? Or has it been shown not to be relevant?

It is not clear how many classes the control group received during the period in which the other group received the intervention. It is advisable to make a brief description of the activities carried out in these physical education classes.

In section 2.4.1 information on methodology and results is included. Each piece of information should be presented in its section for greater clarity.

In L 174, what does it mean that the assessment is carried out following a protocol of 15 sessions?

In the results section, what does the interaction effects between groups and time refer to? What data do the Group x Time columns show?

In the Discussion section, results are included. The information should be structured and located in the corresponding section.

The discussion should place more emphasis on establishing similarities or differences between the current work and other previously published studies, in relation to the characteristics of the sample (age of the children, for example), physical activity program used in the intervention program, follow-up time, etc.

The conclusions should faithfully reflect the data extracted from the study.

The bibliography should be more current; there are no references from the last 5 years.

Thank you very much for your valuable feedback. Below are responses to each of your comments:

  1. Reason for selecting the 8-9 age group
    The 8-9 age group was chosen because this period was critical for acquiring fundamental motor skills (FMS). At this stage, motor interventions were more effective.

  2. Consideration of the etiology of DCD
    The study did not consider the etiology of developmental coordination disorder (DCD), as it focused on assessing the intervention's effect on motor performance in children with DCD.

  3. Number and content of control group classes
    The control group participated in regular physical education classes 2-3 times per week during the intervention period, which included basic physical activities and team sports. A brief description of these activities was added to the methodology section.

  4. Separation of methodology and results in Section 2.4.1
    As you pointed out, the methodology and results needed to be separated. The result data in the methodology section was moved to the results section for clarity.

  5. Meaning of the 15-session protocol
    The “15-session protocol” referred to the standardized Interactive Metronome (IM) assessment, consisting of 15 sessions. This was clarified in the manuscript.

  6. Interaction effects between group and time
    The interaction effects between groups and time indicated that the intervention and control groups experienced different changes over time. The Group x Time columns showed the extent of these changes before and after the intervention. Further elaboration was provided in the results section.

  7. Results mentioned in the discussion section
    As you pointed out, some results were mentioned in the discussion section. These were moved to the results section to ensure a clearer structure and flow in the discussion.

  8. Comparison with previous studies in the discussion
    We agreed that a comparison with previous studies was needed. More discussion on the similarities and differences between this study and previous studies, particularly in terms of age, intervention programs, and follow-up periods, was added.

  9. Reflecting data in the conclusions
    The conclusion was revised to ensure it accurately reflected the findings from the study, emphasizing the key results.

  10. Updating references
    References were updated to include more recent studies from the past five years, ensuring that the literature reflected the most current research.

Your valuable insights significantly contributed to improving the quality of the manuscript. Thank you once again, and we looked forward to your review of the revised manuscript.

Round 2

Reviewer 1 Report

Comments and Suggestions for Authors

The indicated changes have been made appropriately

Reviewer 3 Report

Comments and Suggestions for Authors

Thank you very much for the modifications.